# Climate change increases threat to plant diversity in tropical forests of Central America and southern Mexico

Miguel A. Ortega[1,2]*, Luis Cayuela[3¤], Daniel M. Griffith[4], Angélica Camacho[5], Indiana M. Coronado[6], Rafael F. del Castillo[7], Blanca L. Figueroa-Rangel[8], William Fonseca[9], Cristina Garibaldi[10], Daniel L. Kelly[11], Susan G. Letcher[12], Jorge A. Meave[13], Luis Merino-Martín[3¤], Víctor H. Meza[14], Susana Ochoa-Gaona[15], Miguel Olvera-Vargas[8], Neptalí Ramírez-Marcial[16], Fernando J. Tun-Dzul[17], Mirna Valdez-Hernández[18], Eduardo Velázquez[19], David A. White[20], Guadalupe Williams-Linera[21], Rakan A. Zahawi[22], Jesús Muñoz[23]*

**1** Instituto Mixto de Investigación en Biodiversidad (IMIB-CSIC), Mieres, Spain, **2** Universidad Internacional Menéndez Pelayo, Madrid, Spain, **3** Departamento de Biología y Geología, Física y Química Inorgánica, ESCET, Universidad Rey Juan Carlos, Móstoles, Spain, **4** Departamento de Ciencias Biológicas y Agropecuarias, EcoSs Lab, Universidad Técnica Particular de Loja, Loja, Ecuador, **5** Universidad Intercultural de Chiapas, Corral de Piedra, Mexico, **6** Universidad Nacional Autónoma de Nicaragua, UNAN-León, León, Nicaragua, **7** Instituto Politécnico Nacional, CIIDIR Oaxaca, Santa Cruz Xoxocotlán, Oaxaca, Mexico, **8** Departamento de Ecología y Recursos Naturales, Centro Universitario de la Costa Sur, Universidad de Guadalajara, Autlán de Navarro, Jalisco, Mexico, **9** Universidad Nacional Autónoma de Costa Rica, Santa Lucía, Barva, Heredia, Costa Rica, **10** Departamento de Botánica, Universidad de Panamá, Campus Universitario Ciudad de Panamá, Panamá, República de Panamá, **11** Department of Botany, Trinity College, University of Dublin, Dublin, Ireland, **12** College of the Atlantic, Bar Harbor, Maine, United States of America, **13** Facultad de Ciencias, Universidad Nacional Autónoma de México, Ciudad de México, Mexico, **14** Instituto de Investigación y Servicios Forestales, Universidad Nacional de Costa Rica, Campus Omar Dengo, Heredia, Costa Rica, **15** El Colegio de la Frontera Sur, Unidad Campeche, Lerma, Campeche, Mexico, **16** El Colegio de la Frontera Sur, San Cristóbal de Las Casas, Chiapas, Mexico, **17** Centro de Investigación Científica de Yucatán, Chuburna de Hidalgo, Mérida, Yucatán, Mexico, **18** Herbario, Departamento Conservación de la Biodiversidad, El Colegio de la Frontera Sur, Chetumal, Mexico, **19** Departamento de Producción Vegetal y Recursos Forestales, Instituto Universitario de Gestión Forestal Sostenible, Universidad de Valladolid (Campus de Palencia), Palencia, Spain, **20** Emeritus Faculty, Program in the Environment, Loyola University, New Orleans, New Orleans, Louisiana, United States of America, **21** Instituto de Ecología, A.C. (INECOL), Xalapa, Veracruz, Mexico, **22** Charles Darwin Foundation, Puerto Ayora, Galápagos, Ecuador, **23** Real Jardín Botánico (RJB-CSIC), Madrid, Spain

¤ Current address: Instituto de Investigación en Cambio Global (IIGC-URJC), Universidad Rey Juan Carlos, Móstoles, Spain
* miguel.ortega@csic.es (MO); jmunoz@rjb.csic.es (JM)

**Data Availability Statement:** All relevant data are within the paper and its Supporting Information files.

## Abstract

Global biodiversity is negatively affected by anthropogenic climate change. As species distributions shift due to increasing temperatures and precipitation fluctuations, many species face the risk of extinction. In this study, we explore the expected trend for plant species distributions in Central America and southern Mexico under two alternative Representative Concentration Pathways (RCPs) portraying moderate (RCP4.5) and severe (RCP8.5) increases in greenhouse gas emissions, combined with two species dispersal assumptions (limited and unlimited), for the 2061–2080 climate forecast. Using an ensemble approach employing three techniques to generate species distribution models, we classified 1924 plant species from the region's (sub)tropical forests according to IUCN Red List categories.

**Funding:** The author(s) received no specific funding for this work.

**Competing interests:** The authors have declared that no competing interests exist.

To infer the spatial and taxonomic distribution of species' vulnerability under each scenario, we calculated the proportion of species in a threat category (Vulnerable, Endangered, Critically Endangered) at a pixel resolution of 30 arc seconds and by family. Our results show a high proportion (58–67%) of threatened species among the four experimental scenarios, with the highest proportion under RCP8.5 and limited dispersal. Threatened species were concentrated in montane areas and avoided lowland areas where conditions are likely to be increasingly inhospitable. Annual precipitation and diurnal temperature range were the main drivers of species' relative vulnerability. Our approach identifies strategic montane areas and taxa of conservation concern that merit urgent inclusion in management plans to improve climatic resilience in the Mesoamerican biodiversity hotspot. Such information is necessary to develop policies that prioritize vulnerable elements and mitigate threats to biodiversity under climate change.

## 1. Introduction

Climate change is among the main drivers of biodiversity decline [1, 2], with growing evidence of its impact on species distribution ranges, including migrations, expansions and reductions [3, 4]. Such changes may lead to the reconfiguration of biological communities, resulting in the emergence of new ecological interactions with uncertain consequences [5, 6]. Under current greenhouse gas (GHG) emissions, it is expected that species associated with warmer climates will expand into new regions, while montane and boreal species will experience range contractions or even go extinct [7, 8]. Such changes in distribution could lead to the extinction of 15–37% of species under mid-range climate warming scenarios by 2050 [9]. The capacity of species to survive in the face of climate change will largely depend on their resilience and ability to disperse to suitable habitats [9, 10]. However, the fate of most species has not been evaluated under projections of future warming.

The tropics are thought to be particularly vulnerable to climate change due to historical climatic stability, which has enabled them to host a high number of rare, specialized and range-restricted species that are sensitive to changes [11]. Given that some of the most important biodiversity hotspots are in tropical regions, anticipating the effects of climate change on tropical species has become a conservation priority [12, 13]. Climate change has negative impacts on tropical forests worldwide, increasing the vulnerability of much of their biodiversity [14], even in protected areas [15]. For example, the stress caused by drought events has put large Neotropical forests, such as the Amazon, at risk of collapse [16, 17]. Consequences of climate change in tropical forests include shifts in community composition [18], reductions in species' distribution ranges [19], loss of forest carbon stocks [20], and changes in plant-plant interactions and forest structure [21].

Within the tropics, Central America is considered to be especially susceptible to climate change given anticipated increases in mean temperature, frequency of extreme temperature events, and precipitation variability [22]. Climate change has been predicted to reduce soil moisture availability with direct effects on plant distributions in the region [23], leading to a decline in wet-adapted species and expansion of generalist or dry forest species [24]. Such distributional shifts are driving a reduction in regional species richness, with especially high losses in some moist montane forests [25, 26]. Moreover, Central America and Mexico, along with the Amazon, are home to the largest number of endemic species in the Neotropics [27], which are at high risk of extinction due to climate change [28]. While the effects of climate

change on regional species distributions have been modeled for selected groups of plants, such as those belonging to a single ecosystem [29, 30] or a spectrum of functional types [24], few studies have assessed the impact of climate change on a significant portion of the flora over multiple ecosystems in Central America and southern Mexico [25]. Analysis of community-wide, large-scale patterns is essential to understand the level of risk the regional flora faces and to take measures to avoid mass extinction.

In this study, we evaluate the vulnerability of Central American and Mexican tropical and subtropical forests to climate change by analyzing changes in habitat suitability of a large number of species under future GHG projections. We investigate threats to plant biodiversity by comparing two alternative GHG concentration pathways, corresponding to a moderate and a severe increase in emissions, combined with two assumptions regarding species dispersal. Our experimental framework thus consists of four scenarios in which we address the following questions: (1) Does the degree of threat to plant species vary between climate change scenarios and under different dispersal assumptions? (2) What is the predicted geographic distribution of threatened species under each of the four experimental scenarios? (3) How are threatened species taxonomically distributed? and (4) Which climate variables have the greatest effect on the proportion of species predicted to become threatened under climate change? We hypothesize that the degree of threat to plant species will increase most under the more severe climate change scenario represented by RCP8.5 and limited dispersal, given that dispersal can facilitate species adaptation to climate change. Moreover, species' habitat ranges are expected to contract both in extent and elevation, with shifts occurring towards higher elevations where the impact of climate change is likely to be mitigated by high precipitation and moderate temperatures. The results of this study will allow us to identify vulnerable species and locate priority areas for plant conservation under future warming in this critical biodiversity hotspot.

## 2. Methods

### 2.1. Study area

Our study area comprised the dry and moist (sub)tropical broadleaf forests of southern Mexico and Central America, excluding islands (Fig 1A). The region is considered to be one of

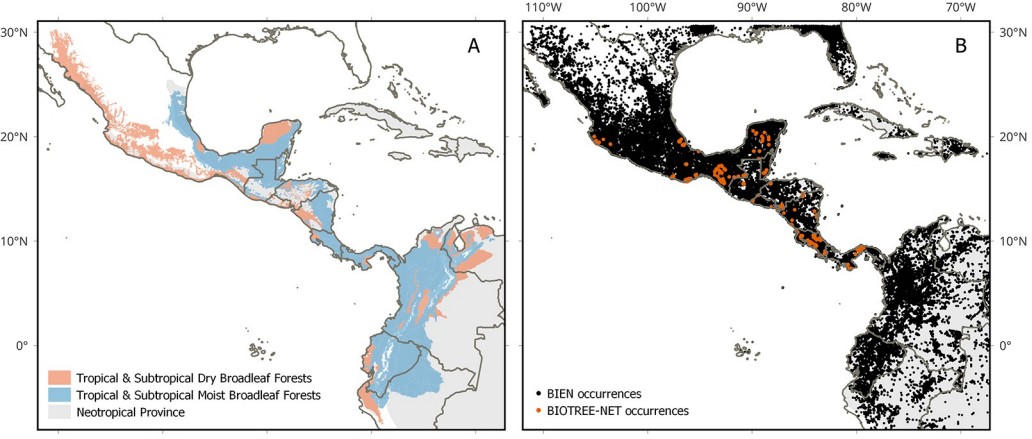

**Fig 1.** (A) Distribution of tropical and subtropical dry and moist broadleaf forests in Mexico, Central America, and the northwestern Andes, based on RESOLVE Ecoregions and Biomes database [61]. (B) Locations of 333,411 plant species occurrences obtained from the BIOTREE-NET [38] and BIEN databases [79] and used to generate species distribution models. Software: QGIS 3.28. Base map source: Natural Earth, available from http://www.naturalearthdata.com/. Ecoregion boundaries obtained from RESOLVE Ecoregions and Biomes database [61], available in https://ecoregions. appspot.com/ under a CC-BY 4.0 license.

the most important biodiversity hotspots on Earth, harboring nearly 2900 endemic plant species [31], but is highly threatened due to ongoing habitat fragmentation and forest degradation [32–34]. To provide more accurate species response curves of the species in relation to environmental variables and reduce biases in the resulting species distribution models, we modeled distributions of the target species over a wider area that included central and northern Mexico as well as the northwestern Andes. The modeling area used was within the limits 110˚ 0' W; 70˚ 0' W; 0˚ 0' N; 30˚ 0' N (Fig 1B). Although modeling distributions across species' full ranges would enable us to capture the complete response curves in relation to environmental variables [35], we selected a smaller area for model calibration to reduce the computational load. This decision was likely to have little impact on the results given that most of the occurrence data for plants from Central America and southern Mexico are located either within this region or also in Colombia or central and northern Mexico [35]. Moreover, inclusion of a larger area for model calibration is likely to have a greater effect on widespread species, which are less impacted by climate change and thus of lesser concern than narrow-ranged species in the context of the present study.

## 2.2. Species data

We focused on tree species due to their essential role in providing ecosystem services and as indicators of multiple diversity components [36, 37], although other woody species such as lianas and palms were also included in the analyses. We obtained a species list from the Tree Biodiversity Network (BIOTREE-NET) [38], which was used to retrieve presence-only data from the Botanical Information and Ecology Network database with the 'BIEN' package [39] in R 3.6.3 [40]. Taxonomic standardization of data was carried out with the 'Taxonstand' package [41]. Individuals not identified to species level (i.e., morphospecies) were discarded from further analyses. To reduce the effects of spatial autocorrelation, we excluded duplicated records within the same pixel (spatial resolution of 30 arc seconds, ~1 km at the Equator). All species with at least 25 unique presences were used for species distribution modeling [42]. In total, we used 333,411 unique occurrences to generate models for 1,924 species in 701 genera and 143 families, using data from both the BIOTREE-NET and BIEN databases (Table 1). The original dataset (S1 Table) contained 410,600 unique occurrences for 2,036 species with more than 25 records, the value considered as the minimum to generate the models. Of these, 112 species were not modeled, either because they did not meet the performance values we set (103, see 2.4 below) or because of errors in the modeling process (9).

## 2.3. Environmental variables

We used climatic variables because of their importance in determining species ranges in tropical forests [43–47]. Climatic variables were obtained from the WorldClim 1.4 dataset [48]. To

**Table 1. Summary of the steps followed to obtain the final dataset used to model plant species distributions.** BIOTREE-NET includes occurrences from southern Mexico and Central America, while BIEN also includes data from Colombia and central and northern Mexico (Fig 1).

| Dataset | Families (N) | Genera (N) | Species (N) | Records (N) |
|---|---|---|---|---|
| Step 1: Original BIOTREE-NET dataset | 154 | 1,163 | 5,148 | 44,650 |
| Step 1.1: BIOTREE-NET records identified to species | 153 | 838 | 2,737 | 39,817 |
| Step 2: BIEN search result matching the BIOTREE-NET species list of Step 1.1 | 153 | 838 | 2,737 | 862,110 |
| Step 3: Merge BIOTREE-NET and BIEN datasets | 153 | 838 | 2,737 | 901,927 |
| Step 4: Remove duplicated occurrences by pixel from the combined dataset | 153 | 838 | 2,737 | 416,717 |
| Step 5: Remove species with less than 25 occurrences or that failed to generate the model | 152 | 809 | 2,036 | 410,600 |
| Step 6: Remove species with less than 25 occurrences | 143 | 701 | 1,924 | 333,411 |

eliminate multicollinearity, we ran a Pearson correlation analysis on 10,000 randomly chosen points across the study area and retained only the most frequently used variable in plant distribution modeling from each variable pair with $r \geq 0.7$ (S2 Table). The variables selected for the models were annual mean temperature, mean diurnal temperature range, temperature seasonality, annual precipitation, and precipitation seasonality. For the 2061–2080 climatic data, we used projections derived from the Global Circulation Model CCSM4 for the Representative Concentration Pathways (RCPs) RCP4.5 and RCP8.5, as defined in the Fifth Assessment Report of the Intergovernmental Panel on Climate Change [49]. RCP4.5 and RCP8.5 correspond to a moderate and severe rise in GHG concentrations, respectively.

## 2.4. Species distribution models (SDMs)

Using the 'biomod2' package [50], we generated SDMs as ensembles of three techniques considered to have high prediction accuracy [51]: generalized boosted models [52], random forests [53], and Maxent [54]. The use of all three techniques was preferred in order to avoid bias that can result from the application of a single modeling algorithm [55]. We used 10,000 randomly selected background points from the same area of the occurrence records to reduce sampling bias [56]. For each species, models were fitted with 70% of the occurrence data and validated with the remaining 30%. To increase robustness of the SDMs, we ran 10 replicates for each modeling technique and used the resulting weighted average as the final ensemble model. Weights were calculated from the true skill statistic (TSS) of each model, using only those with TSS > 0.7. The species list, number of occurrences, and parameters used to evaluate the models are presented in S3 Table. Based on these outputs, we corrected each species' distribution model by applying an exponential decay function of distance from the presence data [57]. Models without this correction represented the 'unlimited dispersal assumption', whereas models corrected by distance represented the 'limited dispersal assumption'. This latter scenario simulates biological and geographical barriers to dispersal by stipulating that the probability of occurrence decreases with distance from the original distribution. Finally, all models were reclassified into presence/absence data using TSS as the threshold.

## 2.5. Species vulnerability

Change in a species' range was obtained as the difference between its present distribution and projected future distribution in each of the four scenarios (i.e., RCP4.5/RCP8.5 and limited/unlimited dispersal). We then classified species as Least Concern (LC), Vulnerable (VU), Endangered (EN), and Critically Endangered (CR) or Extinct (EX) according to the A3 criterion of the IUCN Red List, which is based on an inferred future decline resulting from a reduction in range or habitat quality up to a maximum of 100 years [58].

To investigate whether there were differences in the proportion of each IUCN category between the four scenarios (research question 1), we applied a Pearson's chi-squared test and calculated the p-value based on 2,000 replicates. The proportion of species predicted to fall into any threat category under future projections, hereafter referred to as the *relative vulnerability*, was calculated by both pixel and family, which enabled us to produce geographic (research question 2) and taxonomic (research question 3) distributions of threatened species. To determine which climate variables have the greatest effect on the proportion of threatened species (research question 4), we analyzed the effect of change in climatic variables under the four scenarios, with change defined as the difference between present and future values. To test the statistical importance of the change in each climatic variable on relative vulnerability, which ranges from 0 to 1, we ran 1000 multivariate generalized linear mixed models (GLMMs) with a beta error distribution and logit link function. Beta is a family of continuous probability

distributions defined on the interval [0,1] and is therefore appropriate for this type of response variable. The identity of each individual locality was included as a random factor in the GLMMs, which accounted for the fact that there are 500 random points (see below) that may not represent all possible values in the relative vulnerability distribution.

All GLMMs were fitted using maximum likelihood with the function *glmmTMB* in the R package 'glmmTMB' [59]. For each iteration, we: (1) selected 500 random points; (2) fitted models with all possible variable subsets; and (3) selected the best model according to the Akaike information criterion (AIC) calculated with the R package 'MuMIn' [60]. Weights were assigned to each of the selected models as the relative likelihood of each individual model versus the best model. Variable importance was given by: (1) average of model weights (*wg*) over all models that included each variable; and (2) the number of models containing each variable (N). All analyses were carried out in R 3.6.3 (R Core Team 2020).

## 3. Results

Climate change was projected to have a negative effect on a large proportion of plant species in Central America and southern Mexico. Under the assumption of unlimited dispersal, the percentage of threatened species ranged from 58% (n = 1118) in RCP4.5 to 66% (n = 1274) in RCP8.5, and under limited dispersal, from 59% (n = 1137) in RCP4.5 to 67% (n = 1275) in RCP8.5 (Fig 2). Compared to RCP4.5, the number of threatened species increased by 14% (1.14 factor) in RCP8.5 under both dispersal assumptions ($\chi^2$ = 628.83, *p* = 0.0005). The number of Critically Endangered species increased from 10% (n = 183) in RCP4.5 to 24% (n = 469) in RCP8.5, while species of Least Concern decreased from 42% (n = 804) to 34% (n = 646) between these scenarios. This difference was more pronounced under the assumption of

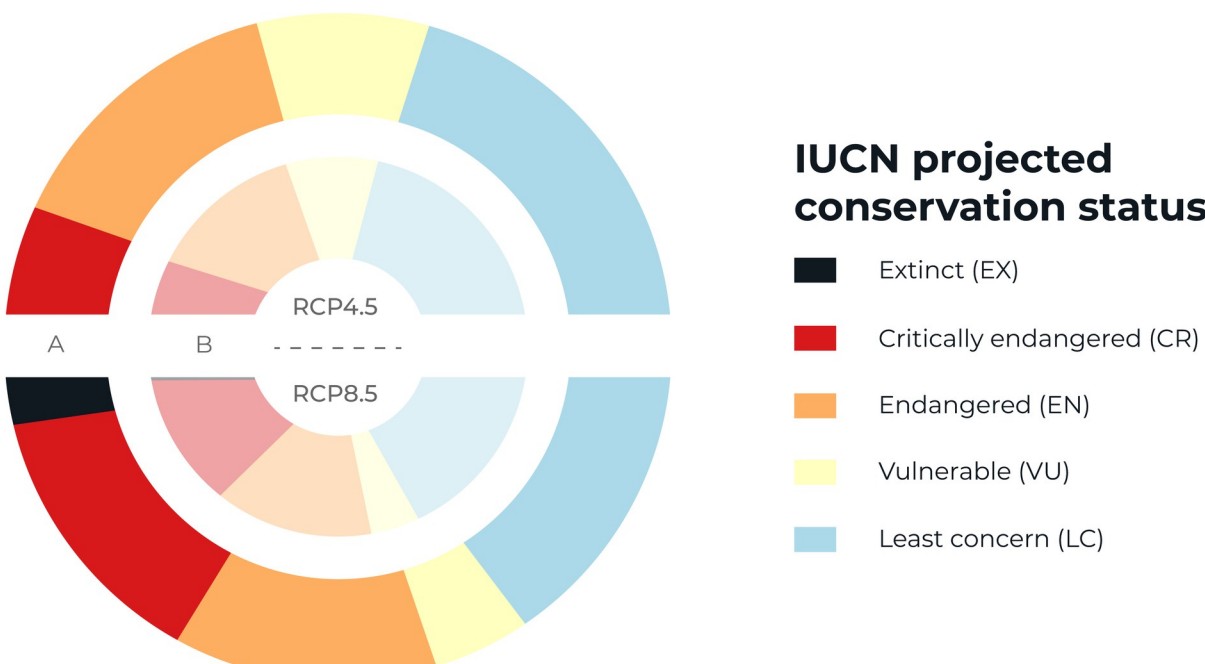

**Fig 2. Projected conservation status of forest species in Central America and southern Mexico under four different climate change scenarios by the year 2070.** We classified species in our dataset (n = 1924) into IUCN Red List categories following the A3 criterion [58]. Different colors represent the percentage of species in each category, which are further separated into two alternative RCPs (top semicircles, RCP4.5; bottom, RCP8.5) and two dispersal assumptions: outer circle (A), limited dispersal; inner circle (B), unlimited dispersal. The complete species dataset is presented in S4 Table).

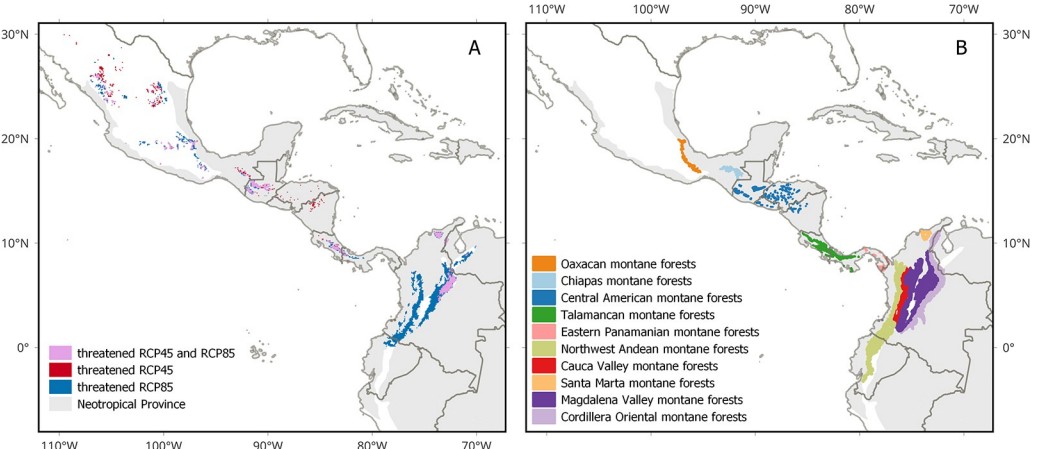

**Fig 3. Projected distribution of threatened species classified as vulnerable, endangered or critically endangered according to the A3 criterion [58].** (A) Projected areas where threatened species exceed 80% of the total number of species per pixel (30 arc seconds) for RCP4.5 and RCP8.5 under the assumption of limited dispersal. (B) The top 10 most threatened ecoregions according to our models, where ecoregions are defined by the RESOLVE Ecoregions and Biomes database [61] (see Table 2 for more details). Software: QGIS 3.28. Base map source: Natural Earth, available from http://www.naturalearthdata.com/. Ecoregion boundaries obtained from RESOLVE Ecoregions and Biomes database [61], available in https://ecoregions.appspot.com/ under a CC-BY 4.0 license.

limited dispersal: 10% (n = 191) compared to 28% (n = 537) of Critically Endangered species and 40% (n = 356) compared to 29% (n = 197) of species of Least Concern between RCP4.5 and RCP8.5, respectively. The number of Endangered and Vulnerable species was similar between limited and unlimited dispersal, while the number of Extinct species increased by 4% (n = 85) in RCP8.5 with limited dispersal relative to RCP4.5 with unlimited dispersal. In general, plant species suffered a reduction in potential distribution ranges (S4 Table).

Threatened species under future climate change were concentrated in montane areas, comprising a larger area overall in RCP8.5 compared to RCP4.5 (Fig 3A). Differences between the two dispersal assumptions were insignificant. The 10 forest ecoregions [61] where vulnerability was highest were all montane forests, namely those of Oaxaca, Chiapas, Central America, Talamanca, Eastern Panama, Northwest Andes, Cauca Valley, Santa Marta, Magdalena Valley, and Cordillera Oriental (Table 2, Fig 3B). In terms of taxonomic distribution, the average

**Table 2. The top 10 most vulnerable forest ecoregions [61] of the study area with the greatest proportion of threatened plant species under future climate change.**
The proportion of threatened species averaged across the pixels composing each ecoregion (mean proportion) and standard deviation (sd), along with average species richness per pixel (richness), are listed by ecoregion. We used all available pixels (n) to calculate the proportion of threatened species in the four experimental scenarios, discarding ecoregions with n < 30 pixels.

| Ecoregion | Mean proportion | sd | Richness | n |
|---|---|---|---|---|
| Talamancan montane forests | 0.78 | 0.10 | 199 | 776 |
| Santa Marta montane forests | 0.78 | 0.16 | 117 | 232 |
| Oaxacan montane forests | 0.75 | 0.10 | 239 | 376 |
| Central American montane forests | 0.68 | 0.21 | 147 | 628 |
| Eastern Panamanian montane forests | 0.65 | 0.11 | 243 | 148 |
| Cauca Valley montane forests | 0.63 | 0.17 | 70 | 1556 |
| Northwest Andean montane forests | 0.62 | 0.19 | 75 | 2904 |
| Chiapas montane forests | 0.62 | 0.18 | 149 | 264 |
| Cordillera Oriental montane forests | 0.59 | 0.25 | 115 | 3211 |
| Magdalena Valley montane forests | 0.55 | 0.28 | 56 | 4908 |

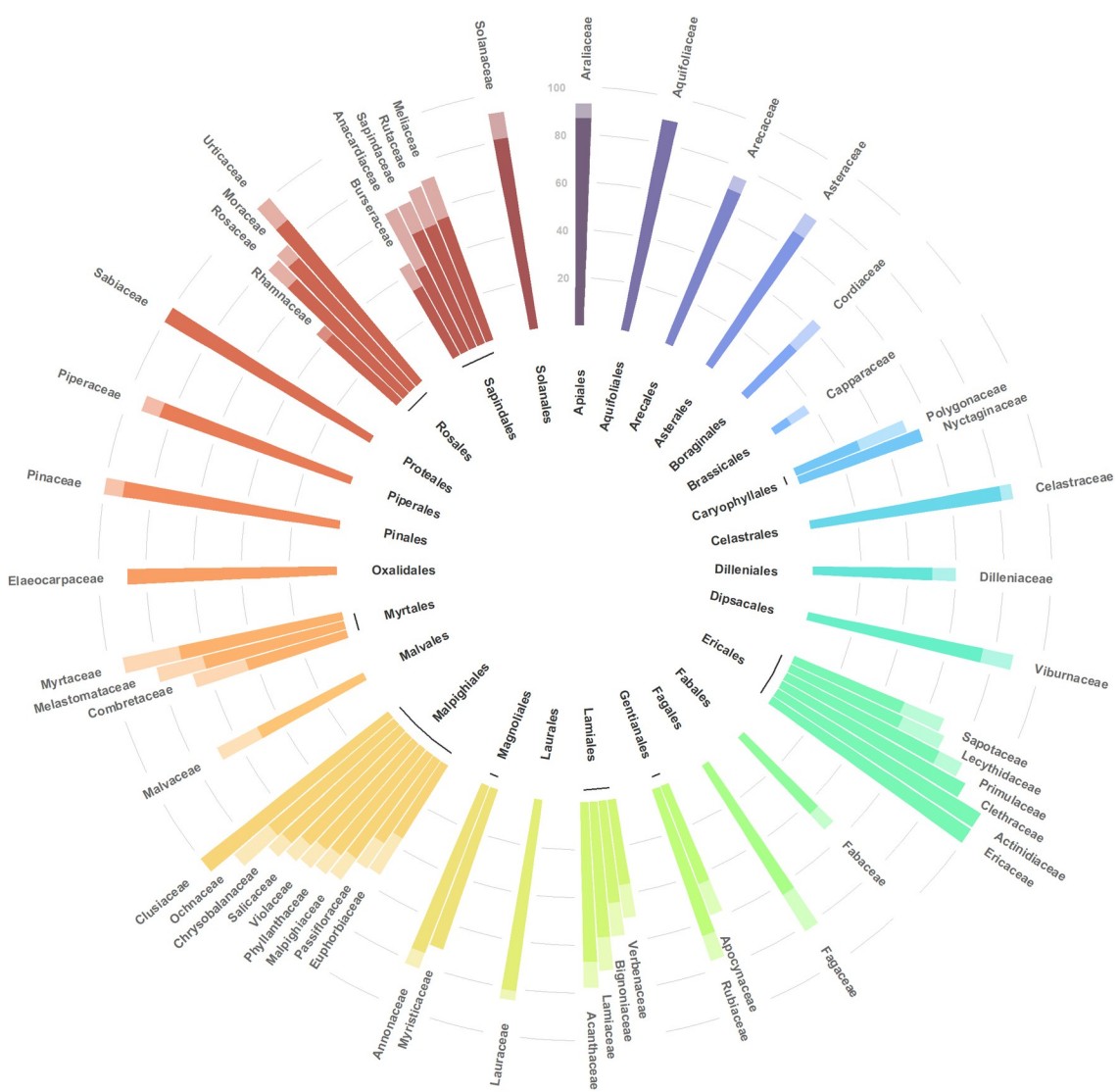

**Fig 4. Percentage of threatened species by family for RCP4.5 (dark bars) and RCP8.5 (dark + light bars) under the assumption of limited dispersal.** Different colors indicate different plant orders. Only families with > 30 species in the dataset are shown; S1 Fig depicts all families, including those with less than 30 species in the dataset.

percentage of threatened species by family (n = 56) was 60% under RCP4.5 and 69% under RCP8.5, rising to 63% and 74% when dispersal was limited, respectively (Fig 4 and S1 Fig). In most families, the number of threatened species was higher under RCP8.5, but differences were not observed between dispersal assumptions. The percentage of threatened species was unequally distributed across families, ranging from below 50% in Cordiaceae, Capparaceae, Verbenaceae, Rhamnaceae and Burseraceae to over 90% in Araliaceae, Actinidiaceae, Ericaceae, Clusiaceae, Myrtaceae, Pinaceae, Sabiaceae, Urticaceae and Solanaceae.

The proportion of species projected to become threatened under climate change was driven primarily by changes in mean diurnal temperature range ($wg = 1$, N = 1000), annual precipitation ($wg = 1$, N = 998), and temperature seasonality ($wg = 1$, N = 795), followed by precipitation seasonality ($wg = 0.1$, N = 239) and annual mean temperature ($wg < 0.01$, N = 124) (Table 3). Mean diurnal temperature range ($z = 5.33$, $p = 0.0000001$) and annual precipitation ($z = 3.008$, $p = 0.0026$) were positively associated with species' relative vulnerability (Table 4).

**Table 3. Results of 1,000 generalized linear mixed model iterations to assess the effect of climatic variables on the proportion of species projected to become threatened under future climate change.** In each run, models with all variable combinations were compared and the best model was selected according to AIC. Variable importance was ranked by: (1) average of model weights (*wg*) over all models that included each variable; and (2) the number of models containing each variable (N).

| | Mean diurnal temperature range | Annual precipitation | Temperature seasonality | Precipitation seasonality | Annual mean temperature |
|---|---|---|---|---|---|
| (1) Average of model weights (*wg*) | 1.00 | 1.00 | 1.00 | 0.10 | < 0.01 |
| (2) No. models containing each variable (N) | 1000 | 998 | 795 | 239 | 124 |

## 4. Discussion

Climate change is projected to threaten approximately 60% of plant species in Central America and southern Mexico due to shifts and reductions in their distributions, reaching as high as 67% under severe GHG emissions and limited dispersal. In addition, three other key findings emerge from our study: (1) For the 2061–2080 climate forecast, the number of threatened species in general and Critically Endangered species in particular will be higher under RCP8.5 than RCP4.5; (2) Dispersal limitation will have little effect on the differences between the RCPs except for Critically Endangered species; and (3) Climate change will disproportionately impact montane areas and plant families associated with these areas. Together, these results underscore the need to avoid moderate to severe climate outcomes and prioritize montane areas and taxa in conservation planning schemes to mitigate threats to the region's unique biodiversity.

The high proportion of threatened species predicted under all four experimental scenarios reveals the vulnerability of the Mesoamerican flora to climate change. Threatened species represented 58–67% of species among the four experimental settings (Fig 2), which resulted from a general reduction in potential distribution ranges (S4 Table). This underscores the need to reduce GHG emissions to below moderate levels, given that even under RCP4.5 and unlimited dispersal, the combined percentage of Endangered and Critically Endangered species was 39%, representing 752 of the 1924 species evaluated. This same trend has been found for European plants [62], but it implies a much larger absolute number of threatened species in the Neotropics given the region's comparatively higher diversity. The results are more concerning under severe GHG emissions, where the percentage of Endangered and Critically Endangered species was projected to be 56% (1,078–1,083 species) with another 4% going extinct by 2070. Both scenarios portend a major disruption to the region's flora, with far-reaching consequences for associated biodiversity and ecosystem function in Mesoamerican forests.

The negative impact of climate change was similar under the two dispersal assumptions except for Critically Endangered species. Climate change is causing populations to track suitable conditions through dispersal mechanisms. However, evidence suggests that the migration rate of some plants is slower than that required to track such conditions at current rates of climate change [10], even for species with high dispersal capacity such as bryophytes [63]. We

**Table 4. Model averaged coefficients of the effect of change in climate variables on the proportion of threatened species under the RCP8.5 and limited dispersal scenario.** Coefficients were estimated using only those models where the variable was included.

| Change in climatic variables | Coefficient | Std. error | z | p |
|---|---|---|---|---|
| Mean diurnal temperature range | 0.0342 | 0.0064 | 5.3300 | < 0.0001 |
| Annual precipitation | 0.0005 | 0.0002 | 3.0080 | 0.0026 |
| Temperature seasonality | -0.0004 | 0.0004 | 1.0750 | 0.2824 |
| Precipitation seasonality | -0.0126 | 0.0109 | 1.1590 | 0.2464 |
| Annual mean temperature | -0.0085 | 0.0075 | 1.1360 | 0.2559 |

modeled the scenario of limited dispersal to reduce over-extrapolating distributions far beyond known occurrence records, thereby simulating the effect of geographic barriers and other factors that restrict dispersal like habitat fragmentation. Although differences between these two assumptions were minor for threatened species generally, the percent change for Critically Endangered species increased by 293% between the most optimistic (RCP4.5, unlimited dispersal) and the most pessimistic (RCP8.5, limited) scenario (Fig 2). This demonstrates a greater degree of threat under severe GHG concentrations and where dispersal barriers are important, as in Central America and southern Mexico where deforestation and habitat fragmentation continue to increase [33]. More research is needed to address which species are most limited by such barriers and develop biological corridors that enable them to track suitable habitat as climate change unfolds [64].

Our results clearly show that the greatest vulnerability corresponds to mountainous regions, with areas of species' vulnerability predicted to be larger under RCP8.5 than RCP4.5. This finding is in agreement with previous studies showing that tropical montane species are particularly sensitive to climate change [16, 17, 23, 24, 65]. For example, mountain areas in Mesoamerica will likely suffer high losses of tree and shrub species richness due to climate change [25]. In our study, the altitudinal risk pattern was consistent across the four scenarios, with montane forests comprising the top 10 ecoregions where vulnerability was highest (Fig 3). Furthermore, the two most important climate variables explaining the proportion of threatened species were diurnal temperature range and annual precipitation, which are associated with montane environments. A recent review showed that the sensitivity of high-elevation zones to temperature changes is due to the high number of endemic species [11]. In the context of climate change, the importance of these regions resides not only in the fact that they harbor more at-risk species, but they will act as refuges for many endemic species, most of which are also at risk due to restricted distributions [28]. The loss of rare and endemic species could result in a significant depletion of ecosystem services, given that rare species represent an important fraction of functional diversity [66].

Most plant families occurring in southern Mexico and Central America contained a high proportion of threatened species; on average 60–74% of the species in diverse families (>30 species) were threatened depending on the RCP/dispersal scenario. The proportion of threatened species was exceptionally high in families associated with Mesoamerican montane forests such as Araliaceae, Clusiaceae, Ericaceae, Myrtaceae and Pinaceae [67], which is unsurprising given that such forests showed the highest vulnerability. These families could lose a large number of species given that many have restricted distributions, which are predicted to contract further as climate change forces them to move up mountains [24]. Identifying which taxa have the highest proportion of species at risk is an important first step for the preservation of genetic diversity. Considering that taxonomy reflects evolutionary history, a further step would be to incorporate phylogenetic information into a research and management framework to conserve taxonomic as well as phylogenetic diversity [68, 69]. The approach presented here provides a useful tool to prioritize taxa for conservation under climate change, particularly in tropical regions, where knowledge on the distribution of many taxa is poorly known [70].

Given that species distributions are also determined by biotic interactions, dispersal ability and evolutionary change, these results should not be interpreted as precise predictions but rather general outcomes to expect under different climate change scenarios [71, 72]. At the working scale of this study, the SDMs generated did not consider species' plasticity, use of microclimatic refuges or biological interactions as factors mediating the effects of climate change. Adaptive plasticity can reduce the negative effects of environmental changes, providing species a degree of resilience in the form of drought resistance, tolerance to thermal extremes, and other traits [73, 74]. Likewise, climate refuges can act as a buffer against local

climatic changes, allowing species with cold temperature requirements, for example, to find refuge on cooler slopes, thus tempering increments in local temperature [75]. On the other hand, the decoupling of plant species from mutualistic partners such as pollinators and seed dispersers, as well as greater exposure to herbivory and disease, can exacerbate the negative effects of climate change [76–78]. Understanding the synergistic effects of these different factors requires targeted research at the population and community level and will be necessary to abate some of the worst impacts of climate change.

In conclusion, a high proportion of plant species in Central America and southern Mexico are predicted to become threatened under moderate to severe climate change. Therefore, safeguard measures should be taken to ensure vulnerable species are protected in strategic areas. While the results between RCPs were not qualitatively different, the proportion of threatened species increased by 14% under RCP8.5 compared to RCP4.5, thus efforts to curb GHG emissions in this century will have a direct effect on the region's biodiversity. Furthermore, addressing deforestation and habitat fragmentation will be crucial to enable plant species to track suitable conditions via dispersal. Montane habitats and associated families, as well as an array of other families, should be prioritized in conservation and restoration planning. Networks of protected areas and sustainable landscapes should be consolidated at the regional level to ensure vulnerable plant species are able to migrate to suitable habitat under climate change.

## Supporting information

**S1 Fig. Percentage of threatened species for all plant families studied.** RCP4.5 (dark bars) and RCP8.5 (dark + light bars) under the assumption of limited dispersal. Different colors indicate different plant orders.
(TIF)

**S1 Table. Dataset of occurrences used in this study.**
(CSV)

**S2 Table. Pearson correlation analysis of 18 variables obtained from Worldclim 1.4 [48] for 10,000 randomly chosen points across the study area.** Cells above the diagonal show the *r* value; cells below the diagonal show the density distribution.
(PDF)

**S3 Table. Name, number of occurrences, and parameters used to evaluate the species distribution model of 1924 plant species evaluated in this study.** TSS, True Skill Statistic.
(XLSX)

**S4 Table. Number of pixels gained or lost and the proportion of change under four experimental scenarios (RCP4.5/RCP8.5 and limited/unlimited dispersal) for 1924 plant species in Central America and southern Mexico.** Status indicates the IUCN threat category.
(XLSX)

## Author Contributions

**Conceptualization:** Miguel A. Ortega, Luis Cayuela, Jesús Muñoz.

**Data curation:** Luis Cayuela, Daniel M. Griffith.

**Formal analysis:** Miguel A. Ortega, Luis Cayuela, Jesús Muñoz.

**Investigation:** Miguel A. Ortega, Luis Cayuela, Daniel M. Griffith, Angélica Camacho, Indiana M. Coronado, Rafael F. del Castillo, Blanca L. Figueroa-Rangel, William Fonseca, Cristina Garibaldi, Daniel L. Kelly, Susan G. Letcher, Jorge A. Meave, Luis Merino-Martín, Víctor H. Meza, Susana Ochoa-Gaona, Miguel Olvera-Vargas, Neptalí Ramírez-Marcial, Fernando J. Tun-Dzul, Mirna Valdez-Hernández, Eduardo Velázquez, David A. White, Guadalupe Williams-Linera, Rakan A. Zahawi, Jesús Muñoz.

**Supervision:** Luis Cayuela, Jesús Muñoz.

**Writing – original draft:** Miguel A. Ortega, Luis Cayuela, Daniel M. Griffith, Jesús Muñoz.

**Writing – review & editing:** Miguel A. Ortega, Luis Cayuela, Daniel M. Griffith, Angélica Camacho, Indiana M. Coronado, Rafael F. del Castillo, Blanca L. Figueroa-Rangel, William Fonseca, Cristina Garibaldi, Daniel L. Kelly, Susan G. Letcher, Jorge A. Meave, Luis Merino-Martín, Víctor H. Meza, Susana Ochoa-Gaona, Miguel Olvera-Vargas, Neptalí Ramírez-Marcial, Fernando J. Tun-Dzul, Mirna Valdez-Hernández, Eduardo Velázquez, David A. White, Guadalupe Williams-Linera, Rakan A. Zahawi, Jesús Muñoz.

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
