## [Decision Letter · Decision Letter 0]

18 Sep 2023

PONE-D-23-22272Climate change increases threat to plant diversity in tropical forests of Central America and southern MexicoPLOS ONE

Dear Dr. Muñoz,

Thank you for submitting your manuscript to PLOS ONE. After careful consideration, we feel that it has merit but does not fully meet PLOS ONE’s publication criteria as it currently stands. Therefore, we invite you to submit a revised version of the manuscript that addresses the points raised during the review process.

Final working title with strong hypotheses.Further development of the plant-herbivory and other existing predatory interactions that likely combine (or not) with climate to influence tree species diversity and distribution.==============================

We look forward to receiving your revised manuscript.

Kind regards,

Charlotte Chibuzor Ndiribe, Ph.D.

Academic Editor

PLOS ONE

3. We note that Figures 1 and 3 in your submission contain [map/satellite] images which may be copyrighted. All PLOS content is published under the Creative Commons Attribution License (CC BY 4.0), which means that the manuscript, images, and Supporting Information files will be freely available online, and any third party is permitted to access, download, copy, distribute, and use these materials in any way, even commercially, with proper attribution. For these reasons, we cannot publish previously copyrighted maps or satellite images created using proprietary data, such as Google software (Google Maps, Street View, and Earth). For more information, see our copyright guidelines: http://journals.plos.org/plosone/s/licenses-and-copyright.

1. You may seek permission from the original copyright holder of Figures 1 and 3 to publish the content specifically under the CC BY 4.0 license. 

4. Please remove your figures from within your manuscript file, leaving only the individual TIFF/EPS image files, uploaded separately. These will be automatically included in the reviewers’ PDF.

5. We notice that your supplementary figures are uploaded with the file type 'Figure'. Please amend the file type to 'Supporting Information'. Please ensure that each Supporting Information file has a legend listed in the manuscript after the references list.

Additional Editor Comments:

Jesús Muñoz et al.,

The study titled "Climate change increases threat to plant diversity in tropical forests of Central America and southern Mexico" appears promising for growing ecology research and outputs from the Neotropics. Also, the main scientific findings of this study indicate average-high numbers of threatened species within the climate scenarios analyzed, suggesting some urgency with more informed species protection, particularly in montane areas. However, I would advise a revision of this manuscript to improve its quality and make it more suitable to this journal. For instance, the title depicts "threat" as if in reference to only one factor.

The study needs to present clear and well structured hypotheses to the questions examined, with justifications. So far, some relevant information is missing with regards to plant-herbivory and other existing predatory interactions that combine or not with climate to influence tree species diversity and distribution. In addition with the reviewer's comments, I strongly recommend the authors to develop the manuscript with accounts for these concerns raised.

Reviewers' comments:

Reviewer's Responses to Questions

**Comments to the Author**

1. Is the manuscript technically sound, and do the data support the conclusions?

Reviewer #1: Yes

Reviewer #2: Yes

2. Has the statistical analysis been performed appropriately and rigorously? 

Reviewer #1: Yes

Reviewer #2: Yes

3. Have the authors made all data underlying the findings in their manuscript fully available?

Reviewer #1: Yes

Reviewer #2: Yes

4. Is the manuscript presented in an intelligible fashion and written in standard English?

Reviewer #1: Yes

Reviewer #2: Yes

5. Review Comments to the Author

Reviewer #1: Such information is important to generate more awareness to protect endangered species and how to mitigate them under new regulations to protect biodiversity. The paper may be as a reference to policy makers.

Reviewer #2: The paper by Muñoz et al. aims at exploring the likely trend for plant species distributions in Central America and southern Mexico under two climate change scenarios namely, Representative Concentration Pathways (RCPs) depicting moderate (RCP4.5) and severe (RCP8.5) increases in greenhouse gas emissions and also under the assumption of limited and unlimited species dispersal for the 2061-2080 climate prediction.

The key findings of the study are that a high proportion (58-67%) of threatened species were found under all-climate scenarios used. The highest proportion of threatened species were found under RCP8.5 and limited dispersal. Threatened species were predominately found in montane areas.

The study results help to priorities areas for conservation and forest management measures to improve the resilience of the biodiversity hotspots of Central America and southern Mexico.

The paper is generally well structured and well written and placed in the proper scientific context. Literature cited is adequate with a balance of old and relatively recent publications from a wide range of authors and Journals. The research questions are clear but there are no corresponding hypotheses to these questions that are being tested.

The methods and analysis are generally scientifically sound, and the results are well interpreted. However, the authors fail to discuss in details other factors such as herbivory and diseases that affect the distribution of tree species. In addition, the paper is about potential effects of climate change (based on the four scenarios or assumptions made) on species distribution, but the title of the paper suggests otherwise. The analysis is not based on the impacts of long terms climate data on the distribution of species but rather how species will respond to assumed scenarios. Therefore, I suggest the title is revised to reflect this reality.

Some minor comments

Line 120: “which confront high extinction risk from climate change”. This phrase is not clear. Please rephrase.

Line 132-133 “The result is an experimental framework consisting of four scenarios where we address the following questions” this statement is a bit confusing; when you talk of results, framework, and questions in the same sentence. Please rephrase.

Line 174: “From a pool of 5,148 species contained in the original dataset, we used 333,411.

unique occurrences to generate models”- do these unique occurrences refer to individuals within a species or records? It is a bit confusing considering that you choose from a pool of 5148 species and ended up with 333411. Please clarify or qualify the statements with “records in the database.”

6. PLOS authors have the option to publish the peer review history of their article (what does this mean?). If published, this will include your full peer review and any attached files.

Reviewer #1: No

Reviewer #2: No

---

## [Author Response · Author response to Decision Letter 0]

4 Dec 2023

•Response: We have reformatted the ms considering PLoS ONE style.

2. In your Data Availability statement, you have not specified where the minimal data set underlying the results described in your manuscript can be found. PLOS defines a study's minimal data set as the underlying data used to reach the conclusions drawn in the manuscript and any additional data required to replicate the reported study findings in their entirety. All PLOS journals require that the minimal data set be made fully available.

Response: We have generated a new supplementary table with all occurrences used to generate the models (Climate change Central American forests Table S1.csv). As this is the first supplementary data in the manuscript, all supplementary tables have been renumbered and uploaded as new datasets.

3. We note that Figures 1 and 3 in your submission contain [map/satellite] images which may be copyrighted. All PLOS content is published under the Creative Commons Attribution License (CC BY 4.0), which means that the manuscript, images, and Supporting Information files will be freely available online, and any third party is permitted to access, download, copy, distribute, and use these materials in any way, even commercially, with proper attribution.

Response: We have added in each Figure the source of the information (all freely available) and the software used to generate them.

4. Please remove your figures from within your manuscript file, leaving only the individual TIFF/EPS image files, uploaded separately.

Response: Done. We have removed all figures from the manuscript and left only the captions in the text.

5. We notice that your supplementary figures are uploaded with the file type 'Figure'. Please amend the file type to 'Supporting Information'. Please ensure that each Supporting Information file has a legend listed in the manuscript after the references list.

Response: Done.

6. Please include captions for your Supporting Information files at the end of your manuscript, and update any in-text citations to match accordingly.

Response: Done.

Response: Done.

Response: Done

Response: Done

Additional Editor Comments:

The study titled "Climate change increases threat to plant diversity in tropical forests of Central America and southern Mexico" appears promising for growing ecology research and outputs from the Neotropics. Also, the main scientific findings of this study indicate average-high numbers of threatened species within the climate scenarios analyzed, suggesting some urgency with more informed species protection, particularly in montane areas. However, I would advise a revision of this manuscript to improve its quality and make it more suitable to this journal. For instance, the title depicts "threat" as if in reference to only one factor.

The study needs to present clear and well structured hypotheses to the questions examined, with justifications. So far, some relevant information is missing with regards to plant-herbivory and other existing predatory interactions that combine or not with climate to influence tree species diversity and distribution. In addition with the reviewer's comments, I strongly recommend the authors to develop the manuscript with accounts for these concerns raised.

Response: Thanks to the Editor for her feedback and constructive comments. Following her advice, we have presented clear hypotheses in the Introduction and discussed the relevance of other threats, like herbivory, that might be influencing tree species distributions in combination with climate change in the Discussion section (see specific responses to Reviewer #2’s comments).

Reviewers' comments:

5. Review Comments to the Author

Reviewer #1: Such information is important to generate more awareness to protect endangered species and how to mitigate them under new regulations to protect biodiversity. The paper may be as a reference to policy makers.

Reviewer #2: The paper by Muñoz et al. aims at exploring the likely trend for plant species distributions in Central America and southern Mexico under two climate change scenarios namely, Representative Concentration Pathways (RCPs) depicting moderate (RCP4.5) and severe (RCP8.5) increases in greenhouse gas emissions and also under the assumption of limited and unlimited species dispersal for the 2061-2080 climate prediction.

The key findings of the study are that a high proportion (58-67%) of threatened species were found under all-climate scenarios used. The highest proportion of threatened species were found under RCP8.5 and limited dispersal. Threatened species were predominately found in montane areas.

The study results help to priorities areas for conservation and forest management measures to improve the resilience of the biodiversity hotspots of Central America and southern Mexico.

The paper is generally well structured and well written and placed in the proper scientific context. Literature cited is adequate with a balance of old and relatively recent publications from a wide range of authors and Journals. The research questions are clear but there are no corresponding hypotheses to these questions that are being tested.

The methods and analysis are generally scientifically sound, and the results are well interpreted. However, the authors fail to discuss in details other factors such as herbivory and diseases that affect the distribution of tree species. In addition, the paper is about potential effects of climate change (based on the four scenarios or assumptions made) on species distribution, but the title of the paper suggests otherwise. The analysis is not based on the impacts of long terms climate data on the distribution of species but rather how species will respond to assumed scenarios. Therefore, I suggest the title is revised to reflect this reality.

Response: We have included a new paragraph at the end of the Introduction to present our hypotheses. Additionally, in the Discussion section, we have incorporated a paragraph to acknowledge that other factors, like herbivory, may also exhibit synergistic interactions with climate change. Climate change has the potential to increase herbivore consumption rates, potentially resulting in heightened foliar and floral damage (Hamann et al., 2020). This, in turn, could increase the threat to certain tropical tree species. Regarding the impact of diseases, it is essential to take into account that tropical forests typically lack truly dominant species, reducing the likelihood of diseases significantly affecting these ecosystems. Furthermore, given that the tree species most susceptible to climate change are often the rarer ones, it becomes even more challenging for them to be impacted by diseases specific to their species (Baltzer et al., 2012; Brenes-Arguedas et al., 2009).

Regarding the title and the emphasis on climate change in our work, we have decided to retain our original title. However, we are open to considering as an alternative title “Potential effects of climate change on tree species distribution threaten plant diversity in tropical forests of Central America and southern Mexico”, if the editor believes this might be more fitting to the contents of the ms.

References in this comment:

Baltzer JL, Davies SJ. Rainfall seasonality and pest pressure as determinants of tropical tree species' distributions. Ecology and Evolution. 2012;2(11):2682-94. Epub 20120927. doi: 10.1002/ece3.383. PubMed PMID: 23170205; PubMed Central PMCID: PMCPMC3501622.

Brenes-Arguedas T, Coley PD, Kursar TA. Pests vs. drought as determinants of plant distribution along a tropical rainfall gradient. Ecology. 2009;90(7):1751–61.

Some minor comments

Line 120: “which confront high extinction risk from climate change”. This phrase is not clear. Please rephrase.

Response: Done.

Line 132-133 “The result is an experimental framework consisting of four scenarios where we address the following questions” this statement is a bit confusing; when you talk of results, framework, and questions in the same sentence. Please rephrase.

Response: Done.

Line 174: “From a pool of 5,148 species contained in the original dataset, we used 333,411 unique occurrences to generate models”- do these unique occurrences refer to individuals within a species or records? It is a bit confusing considering that you choose from a pool of 5148 species and ended up with 333411. Please clarify or qualify the statements with “records in the database.”

Response: Done.

---

## [Editor Report · Decision Letter 1]

15 Jan 2024

Climate change increases threat to plant diversity in tropical forests of Central America and southern Mexico

PONE-D-23-22272R1

Dear Dr. Jesús Muñoz,

We’re pleased to inform you that your manuscript has been judged scientifically suitable for publication and is formally accepted for publication, pending on the implementation of identified ammendments.

Within one week, you’ll receive an e-mail detailing any further required amendments. When these have been addressed, you’ll receive a formal acceptance letter and your manuscript will be scheduled for publication.

Kind regards,

Charlotte Chibuzor Ndiribe, Ph.D.

Academic Editor

PLOS ONE
---

## [Editor Report · Acceptance letter]

20 Feb 2024

PONE-D-23-22272R1 

PLOS ONE

Dear Dr. Muñoz, 

I'm pleased to inform you that your manuscript has been deemed suitable for publication in PLOS ONE. Congratulations! Your manuscript is now being handed over to our production team.

Kind regards, 

on behalf of

Dr. Charlotte Chibuzor Ndiribe 

Academic Editor

PLOS ONE